# Heat Transfer Performance of 3D-Printed Aluminium Flat-Plate Oscillating Heat Pipes for the Thermal Management of LEDs

**DOI:** 10.3390/mi13111949

**Published:** 2022-11-11

**Authors:** Chao Chang, Yaoguang Yang, Lilin Pei, Zhaoyang Han, Xiu Xiao, Yulong Ji

**Affiliations:** Institute of Marine Engineering and Thermal Science, Marine Engineering College, Dalian Maritime University, Dalian 116026, China

**Keywords:** 3D printing, oscillating heat pipe, thermal resistance, heat transfer performance

## Abstract

With the rapid development of electronic technologies towards high integration, high power and miniaturization, thermal management has become an increasingly important issue to guarantee the reliability and service life of electronic devices. The oscillating heat pipe (OHP), which was governed by thermally excited oscillating motion, was considered as a promising technology to dissipate high-density heat and had excellent application prospects in many important industrial processes. A flat-plate OHP, however, was fabricated by traditional welding methods, which were difficult and inefficient, resulting in increasing the cost and wasting the production time. In this work, we adopted a new metal 3D printing technology to develop an aluminum flat-plate OHP, which made it facile to build complex inner channels with high-precision molding at one time. AlSi_10_Mg powders, as raw materials, were selectively melted and solidified to form the container of the flat-plate OHP. The sintered inner surface presented excellent wettability to the working fluid, which facilitated the evaporation of the working fluid. Acetone was chosen as the working fluid, and the filling ratios with a range of 40–70% were loaded into the flat-plate oscillating heat pipe to analyze its effect on heat transfer performance. It was found that the 3D-printed flat-plate OHP with a 60% filling ratio had a better heat transfer performance and a lower thermal resistance, and it was able to work properly in both vertical and horizontal operation modes. The 3D-printed flat-plate OHP had been successfully applied for the thermal management of high-power LEDs, and the results showed that the temperature of LEDs was maintained within 60 °C, and its service life was prolonged.

## 1. Introduction

The rapid development of electronic technology and packaging technology has greatly accelerated the miniaturization and integration of electronic components in recent years. The continuously increasing power density in electronics has led to higher requirements for thermal management technology [1,2]. Due to the superior heat transfer capability, simple construction and low cost, the flat-plate oscillating heat pipe (OHP), as a liquid-vapor heat transfer device, has attracted tremendous interest for its potential applications from both academic and industrial researchers [3]. An oscillating heat pipe is partially filled with the working fluid and typically consists of three sections: an evaporator section, a condenser section and an adiabatic section. In the oscillating heat pipe, a series of vapor plugs and liquid slugs will be naturally formed and distributed along the pipe because of the high surface tension of the working fluid and the small pipe diameter. When heat is added to the outer surface of the evaporator section of the pipe, the vapor plugs begin to extend to push the liquid slugs to the condenser section. Unlike conventional heat pipes, the temperature difference between the condenser section and evaporator section drives the oscillating motion instead of the capillary force generated by the wick structure in the pipe. From this operating principle, it should be noted that the heat transfer process of the flat-plate OHP contains both latent heat and sensible heat, thereby leading to a high thermal conductivity [4,5,6]. The flat-plate OHP has been proven to be particularly effective in thermal management, and it is successfully used in aerospace, electronics, solar energy and many other important industrial fields [7,8,9,10,11].

Since the first oscillating heat pipe was proposed by Akachi in the 1990s [12], many efforts have been made by researchers to explore its operating mechanism and optimize the heat transfer capability in the past few decades. On one hand, many investigations have focused on establishing theoretical models to analyze the flow characteristics of working fluid during its operation and predict the heat transfer performance [13,14,15,16,17]. For example, Lin et al. [18] built a mathematical and physical CFD simulation model to investigate the heat transfer mechanism and predict the heat transfer capability of the OHP. Daimaru et al. [19] developed a numerical model to analyze the thermal performance of the OHP and understand the startup behavior. On the other hand, extensive experiments have been carried out to explore the influence of related factors on the heat transfer performance of the OHP, such as the filling ratio [20,21,22,23], tilt angle [23], working fluid [24,25,26,27], etc. Markal et al. [28] fabricated an OHP which was filled with a mixture of deionized water, methanol and pentane at a ratio of 1:1:1, 1:2:3 and 1:3:2, respectively, and the filling rate was 50%. The experimental results indicated that the mixtures had a better performance than the pure counterparts. Jang et al. [29] designed and fabricated a flat-plate OHP against a thermal graphite sheet for the application of mobile electronic devices. When the inclination angle was 0° and 90°, the thermal resistance of the flat-plate OHP was reduced by 56% and 62%, respectively, compared to that of the thermal graphite sheet. Takawale et al. [30] investigated the thermal performance of a flat-plate OHP and a capillary tube OHP with filling ratios of 40%, 60% and 80% at heat inputs ranging from 20 W to 160 W. The results showed that the flat-plate OHP had a better heat transfer performance with lower thermal resistance compared to the capillary tube OHP. In addition, other researchers have explored the use of OHPs in high-power LEDs, batteries, thermal energy storage, high-power communication equipment and other electronic devices [31,32,33,34,35].

Despite numerous significant investigations having been carried out to improve the thermal performance of the OHPs, it was still a challenge to fabricate flat-plate OHPs in a simple way, especially for the complex internal channels. In general, flat-plate OHPs were usually fabricated by using traditional welding methods, which were difficult and inefficient, resulting in increasing the cost and wasting the production time. 3D printing, known as additive manufacturing, was a technology which was applied to fabricate objects layer-by-layer with different raw materials, such as metal, plastic and ceramics [36,37,38]. Due to the flexible design, rapid prototyping and fast production, 3D printing technology has shown great potential in the automotive industry, shipbuilding, aerospace, etc. 3D printing technology not only made it facile to form products with high-precision molding at one time but also built the surface with different microstructures and improved the surface wettability [39,40]. In this work, we designed and fabricated an aluminum flat-plate OHP by using a selective laser melting (SLM) technology, which was a type of metal 3D printing technology. The sintered aluminum particles not only increased the surface roughness but also presented excellent wettability to acetone, further improving the heat transfer performance of the flat-plate OHP. Furthermore, we investigated the influence of physical parameters, including the filling ratio, input power and inclination angle, on the heat transfer performance of the flat-plate OHP. As a result, the flat-plate OHP with a filling ratio of 60% had the best thermal performance with the lowest thermal resistance. Finally, the developed flat-plate OHP, which was integrated with the LEDs system, was also tested, and the results showed that the flat-plate OHP was able to maintain the normal operation of LEDs and prolonged the serving lifetime.

## 2. Experimental Methods

### 2.1. Materials

AlSi_10_Mg powders were bought from CNPC POWDER Co., Ltd. (Shanghai, China). Acetone was provided by Shanghai Aladdin Reagent Co., Ltd. (Shanghai, China). Ar gas (99.99%) was purchased from Dalian Date Gas Co., Ltd. (Dalian, China). The electric heater (60 mm ∗ 20 mm) was bought from Wujiang Zhongzheng Electric Technology Co., Ltd. (Suzhou, China). Thermal conductive grease was ordered from Shanghai Taipu New Materials Technology Co., Ltd. (Shanghai, China).

### 2.2. Flat-Plate Oscillating Heat Pipes Fabrication

A metal 3D printer (EOS M290) was employed to fabricate the container of the aluminum flat-plate OHP. AlSi_10_Mg particles with a diameter of about 20 µm were used as the raw materials. Figure 1a shows the fabrication process of the flat-plate OHP using a 3D printing technology. Firstly, a thin layer of AlSi_10_Mg powder was evenly spread on the building platform. Then, a high-power laser was turned on and melted and fused the specific part of the powder layer. After the melted part solidified, the building platform would move down from a slight distance, and the surface was covered with another AlSi_10_Mg powder layer by using a recoater. The high-power laser continuously and selectively melted the specific part of this new powder layer. Through the layer-by-layer process, the aluminum powder was finally built to the shape of the desired sample. During the whole process, Ar was severed as a protective gas to avoid the oxidation of the sintered aluminum particles.

The schematic diagram of the flat-plate OHP structure is presented in Figure 1b. As shown, the overall dimensions of the POHP were 90 mm (length) × 60 mm (width) × 5 mm (thickness). The internal channels were composed of seven turns, and the cross section was circular. The distance between the two adjacent internal channels was 2 mm. The channel diameter of the flat-plate OHP was a key factor affecting the flow state of the working fluid and further affecting the heat transfer property of the whole device. Based on previous studies, the channel diameter D should be small enough to ensure that the surface tension overcomes gravity, resulting in the formation of series of liquid plugs and vapor slugs. The Bond number—the ratio between the gravity and surface tension—was usually used to evaluate the theoretical maximum inner diameter of the capillary, and it was calculated by the flowing formula:(1)$$Bo=\frac{g\left(\rho_{l}-\rho_{v}\right)}{\sigma }D^{2}$$
where σ, g, $\rho_{l}$ and $\rho_{v}$ represent the surface tension, gravitational acceleration, working fluid liquid density and working fluid gas density, respectively. When the Bo is equal to 2, the inner diameter D is maximum. In addition, the best range of the OHP inner diameter was calculated by the following formula:(2)$$0.7\sqrt{\frac{\sigma }{g\left(\rho_{l}-\rho_{v}\right)}}\leq D\leq 1.8\sqrt{\frac{\sigma }{g\left(\rho_{l}-\rho_{v}\right)}}$$

In this work, we chose acetone as the working fluid of the flat-plate OHP. According to the physical property of acetone, the channel diameter was designed to be 2 mm, which satisfied Equations (1) and (2). Based on our previous studies [41,42], a filling-back method was used to fill the flat-plate OHP with different filling ratios. The flat-plate OHP was first evacuated by a vacuum air pump to a vacuum below 5 Pa, and the working fluid would be filled into the entire heat pipe by the pressure difference. The part of the working fluid was drawn out, and then the flat-plate OHP was sealed. The 3D-printed OHP was first sealed by a sealing clamp and was then sealed by welding, which could withstand a high temperature of more than 200 °C. We defined the filling ratio as the ratio of the filled working fluid to the entire internal space of the flat-plate OHP.

### 2.3. Measurement Procedure and Data Analysis

The experimental setup for evaluating the heat transfer performance of the 3D-printed flat-plate OHP is presented in Figure 2. As shown, in the evaporator section of the flat-plate OHP, an electric heater with a size of 60 mm ∗ 20 mm was connected to an adjustable DC power to provide input heating power. In the condenser section, a cooling block with a constant temperature of 20 °C was used to absorb the transferred thermal energy. To minimize heat loss, the whole flat-plate OHP was wrapped by a layer of porous thermal insulation materials. In order to analyze the heat transfer capability of the 3D-printed OHP, four T-type thermocouples, which were used to measure the real-time temperature of the flat-plate OHP, were evenly distributed on the evaporator section and the condenser section, respectively.

In the experiment, the input heating power was adjusted from 10 W to 80 W with a step of 10 W each time by using the DC power. At each input power point, the measurement time lasted for 20 min to ensure that the flat-plate OHP reached a steady working state. In addition, thermal resistance (*R*) was an important indicator which had been generally used to evaluate the heat transfer process. The smaller the thermal resistance, the better the performance of the heat pipe. The thermal resistance of the 3D-printed flat-plate OHP was obtained by the following equation:(3)$$T=\frac{1}{N}\sum_{i=1}^{N}T$$
(4)$$R=\frac{\bar{T_{e}}-\bar{T_{c}}}{q}$$
where N is the number of thermocouples in the evaporator section or condenser section, q is the input heating power provided by the DC power supply to the heater from 10 W to 80 W and $\bar{T_{e}}$ and $\bar{T_{c}}$ are the average temperatures of the evaporator section and the condenser section, respectively.

## 3. Results and Discussion

In order to characterize the wettability of the inner surface in the 3D-printed flat-plate OHP, we cut the pipe in half lengthwise. Figure 3a shows the micromorphology of the inner surface of the 3D-printed flat-plate OHP with different magnifications, which were characterized by scanning electron microscopy (SEM). As shown, the 3D-printed flat-plate OHP had a rough inner surface, and numerous granular bulges were distributed on the surface. Such structure was attributed to the selective laser melting technology, which melted and fused aluminum powders by a high-power laser. The sintered aluminum powders formed such a rough surface.

Figure 3b displays the wettability of the inner surface by using a contact angle measurement instrument. The result showed that the contact angle of acetone was almost zero. When an acetone droplet attached to the inner surface of the 3D-printed flat heat pipe, it quickly spread on the inner surface of the flat-plate OHP in less than 1 s. The sintered rough surface presented excellent wettability to the working fluid and provided a strong capillary capability, which facilitated the evaporation of the working fluid.

Figure 4a shows the temperature variation of the flat-plate OHP with a 60% filling ratio over time when the input heating power increased from 10 W to 80 W in a vertical operation mode. When the input power added to the evaporator section of the flat-plate OHP increased from 10 W to 30 W, all measured temperatures rose rapidly until the OHP reached a steady state. In this stage, no temperature fluctuation was observed, meaning that the heat was mainly absorbed by the working fluid. When the heat power was increased to 40 W, all the measured temperatures begun to fluctuate, which indicated the occurrence of the startup. The measured maximum temperature on the evaporator section increased from 58.7 °C at 40 W to 62.5 °C at 50 W, 63.6 °C at 60 W, 68.3 °C at 70 W and 70.4 °C at 80 W. When the flat-plate OHP reached a steady state, the thermal resistance *R* could be obtained according to Equations (3) and (4). 

The amount of working fluid played a significant role in the heat transfer process of the flat-plate OHP. In general, when the flat-plate OHP was filled with too much working fluid, the liquid plugs would impede the oscillating motion, leading to a poor heat transfer performance. In contrast, if the amount of working fluid in the heat pipe was too low, there would not be enough liquid at the evaporator section to absorb thermal energy to generate the driving force for the oscillating motion. Therefore, there should be an optimized filling ratio existing in the flat-plate OHP. Figure 4b presented the evolution of the thermal resistance of the flat-plate OHP filled with different filling ratios ranging from 40% to 50%, 60% and 70% at various input powers. As shown, it should be noted that the thermal resistance of the flat-plate OHP decreased with higher input powers. The 3D-printed flat-plate OHP with a filling ratio of 60% presented the better heat transfer performance with the lower thermal resistance compared to the other three heat pipes with filling ratios of 40%, 50% and 70%. When the input power was 80 W, the thermal resistance of the flat-plate OHP with a 60% filling ratio was less than 0.5 °C/W. According to the experimental results, it can be clearly seen that 60% is the optimal filling ratio for this 3D-printed flat-plate OHP. When the OHP was loaded with 60% working fluid, there was enough space in the pipe for the generation of vapor bubbles and liquid plugs, and a fairly uniform distribution of liquid and vapor could be formed in the OHP. When heat was added on the surface of the OHP, a large pressure imbalance between the adjacent tube could be easily generated. During the operation of the OHP, the OHP with a filling ratio of 60% had enough working fluid to complete the circulation and heat transfer from the evaporator section to the condenser section. A total of 60% of the working fluid could avoid the OHP to dry out, thereby resulting in an excellent thermal performance.

To further verify the application in practice, we evaluated the heat transfer capability of the 3D-printed flat-plate OHP when it was horizontally placed to eliminate the effect of gravity. Figure 4c presents the variation in the thermal resistance of a 3D-printed flat-plate OHP with a 60% filling ratio in the horizontal operation mode. As shown, when the filling ratio of the flat-plate OHP was 60%, the thermal resistance was gradually decreased as the input power increased from 40 W to 80 W. In addition, it was obvious that when the input heating power was increased from 40 W to 80 W, the 3D-printed flat-plate OHP presented a good thermal performance and low thermal resistance regardless of whether it was in the vertical or horizontal operation modes.

Finally, because of the excellent heat transfer capability of the flat-plate OHP, we explored its practical application for the thermal management of high-power LEDs. As shown in Figure 5a, an LED chip that generated a thermal power of 40 W was placed at the evaporator section of the flat-plate OHP, and a cooling block was connected to the condenser section of the flat-plate OHP at a constant temperature of 20 °C. The thermal energy produced by the LEDs was first transferred through the developed flat-plate OHP and was then dissipated by the cooling water. Additionally, a T-type thermocouple was attached to the back of the LEDs in order to measure and record the real-time temperature variation of the LEDs. 

Figure 5b presents the temperature variation of the LEDs over time. As shown, when the LEDs were lit up without the operation of the flat-plate OHP, the temperature of the LEDs would quickly reach more than 170 °C within 1 min, leading to the damage of the LEDs. On the contrary, when the LEDs were connected to the flat-plate OHP, the temperature would quickly reach a steady working state at 60 °C, thus prolonging the service life of the LEDs. Additionally, we also investigated the effect of gravity on the heat transfer capability of the flat-plate OHP. When the 3D-printed flat-plate OHP was placed horizontally, the temperature of the LEDs was maintained at 63 °C, which was slightly higher than that in the vertical operation mode. Figure 5c shows that the LEDs were lit up when they were connected to the evaporator section of the 3D-printed flat-plate OHP. Therefore, the results showed that the LEDs were able to function properly with the assistance of the flat-plate OHP.

## 4. Conclusions

In summary, we successfully developed an aluminum flat plate oscillating heat pipe (90 mm × 60 mm × 5 mm) by using 3D printing technology. This 3D printing technology realized the manufacture of an integrated molding with complex internal channels produced in a short production cycle time; meanwhile, the sintered surface provided enough capillary capability, which facilitated the liquid vaporization at the evaporator section of the flat-plate OHP. Acetone was used as the working fluid for the flat-plate OHP, and the influence of different filling ratios, heating powers and tilt angles on the thermal performance of the flat-plate OHP was analyzed. The experimental results showed that the flat-plate OHP with a filling ratio of 60% had the best thermal performance with the lowest thermal resistance. When the input power was 80 W, the thermal resistance of the flat-plate OHP was less than 0.5 °C/W. The 3D-printed flat-plate OHP could work properly regardless of whether it was placed vertically or horizontally. Moreover, the 3D-printed flat-plate OHP was successfully applied in the thermal management of LEDs. It not only maintained the normal operation of LEDs at 60 °C but also prolonged their service life. Besides high-power LEDs, the 3D-printed flat-plate OHP is expected to be employed in high-power CUP, miniaturized electronics, highly integrated batteries and other systems that involve the requirement of heat dissipation.

## Figures and Tables

**Figure 1 micromachines-13-01949-f001:**
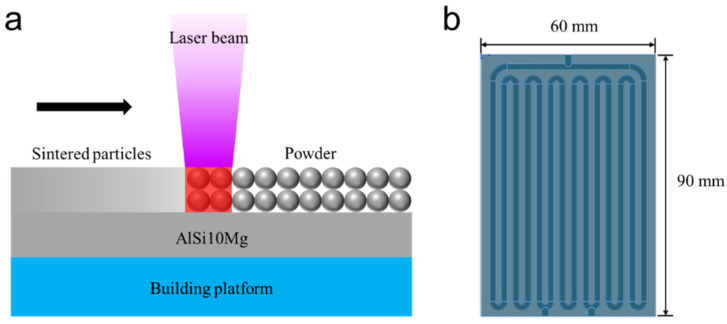
(**a**) Schematic of the 3D printing fabrication process. (**b**) Schematic of the structure of the flat-plate oscillating heat pipe.

**Figure 2 micromachines-13-01949-f002:**
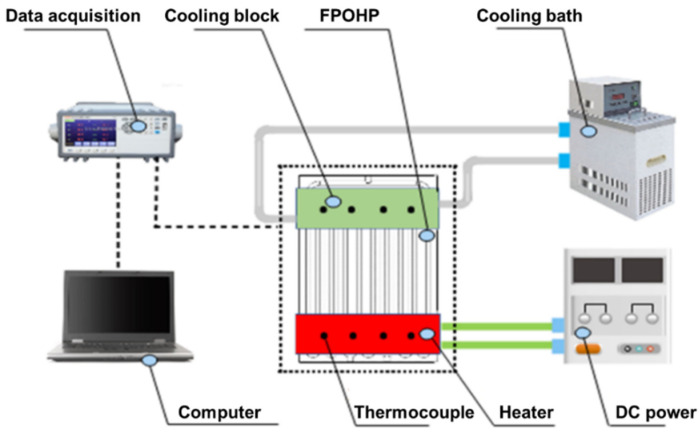
Schematic diagram of the experimental setup.

**Figure 3 micromachines-13-01949-f003:**
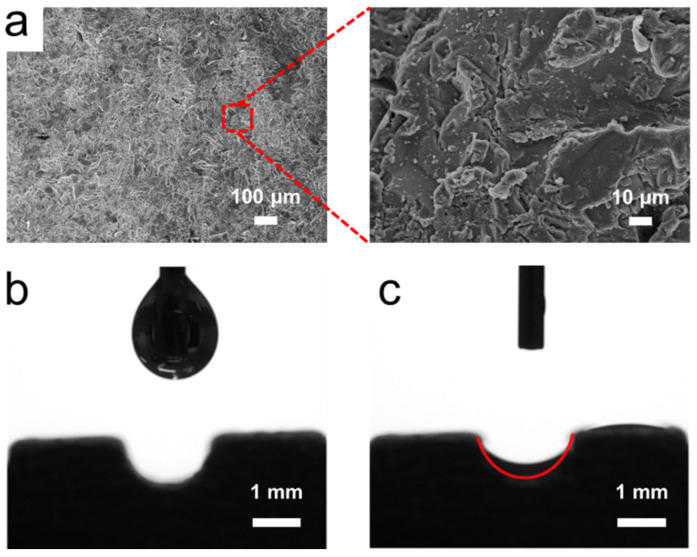
(**a**) SEM images of the inner surface of the 3D-printed flat-plate OHP. Optical images showing an acetone drop (**b**) before and (**c**) after attaching the inner surface.

**Figure 4 micromachines-13-01949-f004:**
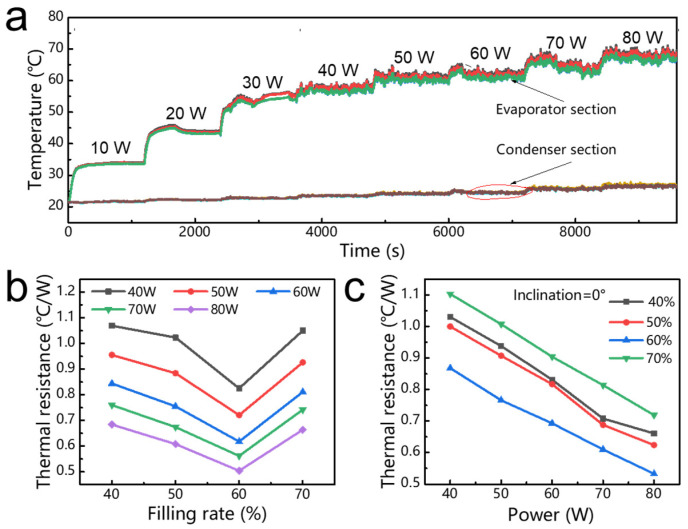
(**a**) Temperature evolution of the flat-plate OHP over time. (**b**) Thermal resistance variation of the flat-plate OHP with different filling ratios under different heating power inputs (in the vertical operation mode). (**c**) Thermal resistance variation of the flat-plate OHP under different heating power inputs (in the horizontal operation mode).

**Figure 5 micromachines-13-01949-f005:**
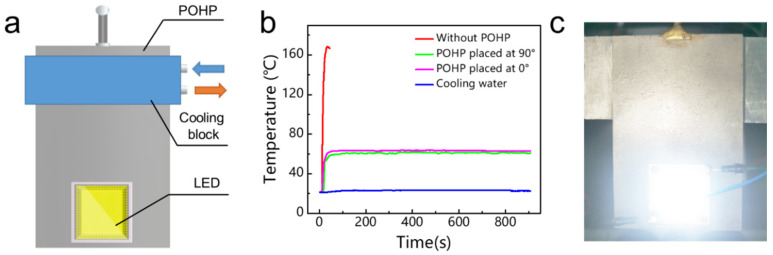
(**a**) Schematic diagram of the experimental setup. (**b**) Temperature variation of the LEDs over time. (**c**) An optical photograph showing that LEDs can light up with the assistance of the flat-plate OHP.

## Data Availability

The data presented in this study are available on request from the corresponding author.

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
