# Peer review of "Heat Transfer Performance of 3D-Printed Aluminium Flat-Plate Oscillating Heat Pipes for the Thermal Management of LEDs"

_micromachines, 2022, doi:10.3390/mi13111949_

Round 1
Reviewer 1 Report
This paper illustrates an oscillating heat pipe (OHP) manufactured by 3D-print. The authors experimentally investigated the heat transfer performance OHP with acetone as the working fluid with different filling ratio. It is a novel method for the manufacture of the OHP. Some comments are as followed:
1. What is the main advantage for the 3d-printed OHP compared with the normal OHP?
2. How is the sealing performance of the 3d-printed OHP. It is very important for the long and stable operation of the OHP.
3. It is better to investigate the startup performance of the OHP.
4. The temperature evolution during the startup and steady stage should be investigated in detail.
5. Both for the vertical and horizontal heating mode, 60% filling ratio is best. The authors should add some explanations the reason.
Author Response
This paper illustrates an oscillating heat pipe (OHP) manufactured by 3D-print. The authors experimentally investigated the heat transfer performance OHP with acetone as the working fluid with different filling ratio. It is a novel method for the manufacture of the OHP. Some comments are as followed:
- What is the main advantage for the 3d-printed OHP compared with the normal OHP?
Reply: Thanks for the comment. Generally, a conventional oscillating heat pipe was made by milling grooves on the bottom plate, and then the bottom plate and the cover plate were welded together by solder. This method was inefficient and leakage between the adjacent channels was easy to occur when the heat flux was high. In this work, we adopted a 3D metal printing technology to fabricate OHPs, which made it facile to build complex inner channels with high precision molding at one time. In addition, the 3D printing technology can form a rough surface with different microstructures, which improved the surface wettability. The sintered aluminum particles not only increased the surface roughness, but also presented excellent wettability to acetone, further improving the thermal performance of the flat-plate OHP.
- How is the sealing performance of the 3d-printed OHP. It is very important for the long and stable operation of the OHP.
Reply: Thanks for the comment. In this work, the 3D printed OHP was first sealed by a sealing clamp, and then was sealed by welding, which could withstand a high temperature of more than 200 ℃. During the experiments, the maximum heating power input was 80 W, and the corresponding maximum temperature did not exceed 100 ℃. Therefore, the sealing performance of the 3D printed OHP can meet the experimental requirements. In our previous work, we also used the same method to fabricate the OHP, and the results showed the developed OHP has excellence thermal performance and stability as follows:
- Ji, Y.; Ma, H.; Su, F.; Wang, G. Particle size effect on heat transfer performance in an oscillating heat pipe. Therm Fluid Sci. 2011, 35, 724-727.
- Yu, C.; Ji, Y.; Li, Y.; Liu, Z.; Chu, L.; Kuang, H.; Wang, Z. A three-dimensional oscillating heat pipe filled with liquid metal and ammonia for high-power and high-heat-flux dissipation. J. Heat Mass Transfer 2022, 194.
- Chang, C.; Han, Z.; He, X.; Wang, Z.; Ji, Y. 3D printed aluminum flat heat pipes with micro grooves for efficient thermal management of high power LEDs. Sci Rep 2021, 11, 8255.
In the revised manuscript, we added more description on the sealing method as follows:
The 3D printed OHP was first sealed by a sealing clamp, and then was sealed by welding, which could withstand a high temperature of more than 200 ℃.
- It is better to investigate the startup performance of the OHP.
Reply: Thanks for the comment. Yes, the startup performance was very important to the OHP. In the revised manuscript, we added description on the startup performance of the OHP as seen the reply of Question 4.
- The temperature evolution during the startup and steady stage should be investigated in detail.
Reply: Thanks for the comment. In the revised manuscript, we added more description the temperature evolution during the startup and steady stage as follows:
When the input power added to the evaporator section of the flat-plate OHP increased from 10 W to 30 W, all measured temperatures rose rapidly until the OHP reached a steady state. In this stage, no temperature fluctuation was observed, meaning that the heat was mainly absorbed by the working fluid. When the heat power was increased to 40 W, all the measured temperature begun to fluctuate, which indicated the occurrence of startup. The measured maximum temperature on the evaporator section increased from 58.7 oC at 40 W, to 62.5 oC at 50 W, 63.6 oC at 60 W, 68.3 oC at 70 W, and 70.4 oC at 80 W.
- Both for the vertical and horizontal heating mode, 60% filling ratio is best. The authors should add some explanations the reason.
Reply: Thanks for the comment. Best filling ratio is still an important research direction of the OHP. Each OHP has a different case-by-case experimental setup, and the best filling ratio was usually determined by the properties of the working fluid, operating temperature range, construction of the OHP, surface enhancements, etc. In previous study, many researchers found the best filling ratio range was 50%-60% (Inter. J. Therm. Sci., 2018, 134: 258-268; Frontiers Energ. Res., 2022: 147; Mater. Today: Proceedings, 2018, 5(10): 22229-22236). Our experimental results showed the best liquid filling rate was 60%. When the OHP loaded with 60% working fluid, there was enough space in the pipe for the generation of vapor bubbles and liquid plugs and a fairly uniform distribution of liquid and vapor could be formed in the OHP. When heat was added on the surface of the OHP, a large pressure imbalance between adjacent tube could be easily generated. During the operation of the OHP, the OHP with a filling ratio of 60% had enough working fluid to complete the circulation and heat transfer from the evaporator section to the condenser section. 60% of the working fluid could avoid the OHP to dry out, thereby resulting in an excellent thermal performance. In the revised manuscript, we added some explanations the reason as follows:
When the OHP loaded with 60% working fluid, there was enough space in the pipe for the generation of vapor bubbles and liquid plugs and a fairly uniform distribution of liquid and vapor could be formed in the OHP. When heat was added on the surface of the OHP, a large pressure imbalance between adjacent tube could be easily generated. During the operation of the OHP, the OHP with a filling ratio of 60% had enough working fluid to complete the circulation and heat transfer from the evaporator section to the condenser section. 60% of the working fluid could avoid the OHP to dry out, thereby resulting in an excellent thermal performance.

Reviewer 2 Report
Authors have studied the Heat transfer performance of 3D-printed aluminium flat-plate 2 oscillating heat pipes for thermal management of LEDs. I my opinion paper is well structured and have authentic results. However, few of my comments are given below.
Novelty of the paper should be included in abstract base on the literature gap.
Improve the figure 4. Data is not visible.
Some quatitative results should be included in conclusion
Author Response
Authors have studied the Heat transfer performance of 3D-printed aluminium flat-plate 2 oscillating heat pipes for thermal management of LEDs. I my opinion paper is well structured and have authentic results. However, few of my comments are given below.
Thanks for the encouraging comments and helpful suggestions. In the revised manuscript, we revised the abstract and conclusion, and modified Figure 4. Below are our point-to-point responses to the reviewers’ comments.
Novelty of the paper should be included in abstract base on the literature gap.
Reply: Thanks for the comment. In the revised manuscript, we added the novelty of the paper in abstract section.
Improve the figure 4. Data is not visible.
Reply: Thanks for the comment. In the revised manuscript, we have revised the Figure 4.
Some quatitative results should be included in conclusion。
Reply: Thanks for the comment. In the revised manuscript, we added some quantitative results in conclusion.